# The Feelings of Nursing Students during the COVID-19 Confinement: Narrative-Based Nursing and Poetry-of-Care Perspectives

**DOI:** 10.3390/ijerph192113919

**Published:** 2022-10-26

**Authors:** José Siles, Elena Andina-Díaz, Carmen Solano-Ruíz

**Affiliations:** 1Nursing and Culture of Care Research Group (EYCC), Nursing Department, University of Alicante, 03690 San Vicente del Raspeig, Spain; 2Nursing and Culture of Care Research Group (EYCC), Nursing Department, University of León, 24004 León, Spain

**Keywords:** nursing, narrative-based nursing, poetry of care, COVID-19, aesthetic nursing, qualitative research

## Abstract

(1) Background: Experiences involve feelings, which, in turn, produce meaning that can become a subjectively lived experience. Therefore, the study of experiences and feelings is essential. Introduction: We examined the role of narrative-based nursing (NBN) and the poetry of care (PC). Objective: To reflect upon the emotions and feelings experienced by nursing students during confinement induced by COVID-19. (2) Methods: This is a qualitative study with a focus on reflexive anthropology, NBN, and PC. Setting and participants: The non-probabilistic sample of incidental, casual, or accessibility type. It consists of 198 students completing their first degree in nursing (the academic year of 2019-2020) of the University of Alicante. (3) Results: Three main categories were considered in the research: For the ‘first day’, 21 subcategories were identified, and uncertainty was the most frequently noted feeling. For the ‘most significant day’, 22 subcategories were found, with the explosion of feelings being the most frequent. For the ‘last day’, 15 subcategories were recorded, with the feeling of relief being the most common. Conclusions: The NBN and PC are relevant therapeutic tools that facilitate reflection and promote awareness of feelings.

## 1. Introduction

The lack of studies on the emotions and feelings that occur during nurse–patient interactions is the justification for this research. Emotions and feelings affect nurses on both professional and personal levels, and a lack of awareness of them can lead to undesirable situations: Demotivation, stress, lack of attention, and even burnout. This study focuses on assessing the feelings experienced by nursing students during confinement during the COVID-19 pandemic. Aesthetics, following Kant in his “Critique of Judgement”, is the science that deals with the systematic study of the origin of feelings and, likewise, the way of materialising them through artistic activity [1]. 

Emotions and feelings were not the subject of research for a long time. The priorities of a neo-positivist paradigm were disinterested in a subject characterized by subjectivity and complexity [2]. Sociopoetics is an attempt to objectify the study of feelings by using intersubjectivity [3]. On the other hand, it is necessary to clarify the language and meanings involved in emotions and feelings. The boundary between emotions and feelings is very permeable and, although there are different trends in this respect, this study assumed that emotions constitute adaptive reactions to situations that often occur in nursing work. Emotions are inherently human behavioural and physiological adjustments that are not usually expressed voluntarily or consciously [4,5]. 

When emotions mature and are consciously expressed, they evolve into feelings, which have been socially constructed [6,7]. In this process of the crystallisation of feelings from emotions, there is an interaction between the intimate and personal sphere of the individual (habitus) with the environment and the immediate social structure (field) [8,9]. The habitus represents the socialization of subjectivity through which intersubjectivity is constructed, giving rise to an idea of normality or an unwritten norm that regulates what individuals should feel, but which can also serve to objectify subjectivity through sociopoetics [3,7,10]. 

Feelings are the real emotions that arise from being involved in something. This something can be another human being, a concept, oneself, a problem, or a situation [5,6]. Poetry as a tool for expressing feelings can help to identify the essential and transcendent aspects of experiences and appreciate them from an ethical perspective [1,11]. Other studies argue that disruptive contexts, such as pandemics, facilitate the reflection and expression of feelings through PC [12]. In this respect, Ricoeur describes the use of writing to re-establish the equilibrium lost after a disruptive event [13]. 

This embodiment of feelings facilitates practical reflection on our experiences [1,2,10,14,15]; or, in other words, reflexive anthropology through a process of enquiry into the consciousness of socially constructed feelings [8,9]. This functionality of poetry has been employed to alleviate the intensity of hurtful feelings or as therapy to relieve stress or depression in nurses [16,17,18,19]. Several studies report that NBN and PC serve to enhance reflection, self-reflection, and hetero-reflection [19,20,21,22,23,24]. NBN consists, on the one hand, in the consideration of the patient’s subjectivity as integrated evidence in their narratives about health–illness processes and, on the other hand, in the understanding and analysis of the underlying meanings of these narratives, contributing to the humanisation of care [15]. PC is the expression of the feelings that arise during the interaction between patients and nurses [14].

**Hypothesis.** 
*The combination of methods such as narrative-based nursing and care poetry is a tool that contributes to the self-awareness of the feelings experienced by the students during the pandemic. On the other hand, the research question guiding this study is the following: What feelings did students experience during the COVID-19 pandemic?*


## 2. Materials and Methods

This is a qualitative study of a socio-critical nature [15] with a focus on reflexive anthropology [9], Narrative-Based Nursing and the poetry of care. It was carried out at the University of Alicante. In total, 198 first-year nursing students participated in the study. We used the Coreq checklist to report the study. For data analysis, we used the QCAmap [25]. We used an approach from reflective anthropology that employs methods such as NBN and PC to facilitate self-knowledge and awareness of experiences and feelings [8,9]. 

The EBN was to facilitate the students’ prose writing of their experiences in order for them to briefly and concisely describe three of their experiences during the confinement caused by the COVID-19 pandemic in narrative form: (a) A narrative about the first day of confinement; (b) a narrative focused on the most significant day during the confinement; and (c) a narrative about the last day of confinement. 

The PC was used to transform the narratives previously written by the students into short poems (4–8 verses) in order to let feelings emerge.

The data were collected from April to June 2020 through a document inspired by NBN and PC. The nursing department professor and director of the Culture of Care research group conducted the data collection process. The students knew the professor because he was the coordinator of the subject chosen as the research setting. Researchers explained the study’s characteristics and objectives before starting it. The data collection document (inspired by the NBN and PC) took one and a half hours to complete. Through a Reading workshop, the narratives and poems were read and discussed. The initial paradigm is the socio-critical one, with the aim of giving voice to and activating the participation of students in the enquiry into emotions and feelings [10]. The theoretical referential was based on Bordieu and Wacquant’s reflexive anthropology assumptions [8,9], considering the habitus and the field (the disruptive scenarios) as tools for analysing the data. Through the habitus, a system of stable references is constructed that serves to overcome the dichotomy between objectivism and subjectivism (the duality between individual and society) and facilitates the analysis of the processes of internalization of the external and externalization of the internal; in the context of a pandemic, the habitus makes it possible to assess the impact of society on people who are experiencing a pandemic crisis and who adopt the classifications that guide their ways of valuing, perceiving, and acting. 

Sample: The non-probabilistic sample of incidental, casual, or accessibility type. It consists of 198 students in their 1st degree in nursing (academic year of 2019–2020) of the University of Alicante. Inclusion criteria: (a) Students enrolled in the subject “fundamentals of nursing” and (b) students who were duly informed of the characteristics and objectives of the study and signed the informed consent form. 

The data collection and analysis technique was performed through a document based on NBN and CP principles.

“NBN-PC Seminar: Feelings during the COVID-19 pandemic” was developed in three phases: (a)A preliminary or “threshold” process consisting of writing a reflection and narrative description of an experience at three moments of confinement: On the first day, the most significant day, and the last day [15].(b)Transthreshold phase: The students elaborated on poems that reflected their feelings analysing previous narratives [14].(c)In the third phase, the narratives and poems were read and debated.

QCAmap software facilitated the organisation and analysis of the data [25]. The starting point was a general category: The first day of confinement during the COVID-19 pandemic. All authors coded the data. The text was segmented/coded by keywords relating to the categorised sentiments in a non-exclusionary way. Saturation occurred well before the end of the analysis process given the unusual sample size for qualitative studies. However, it was decided to integrate the whole population into the study given the pedagogical characteristics (stimulation of creative thinking and practical reflection) and its therapeutic potential in the context of the pandemic. This study followed all the ethical principles contained in the Declaration of Helsinki. All participants signed the informed consent form after receiving clear and truthful information regarding the characteristics and objectives of the study and their rights as citizens involved in the research process.

## 3. Results

Regarding the performance of the PC and the NBN, the students stated the following: 98% (*n* = 185) felt that it increased their understanding of lived reality in confinement, 88% (*n* = 166) stated that it contributed to self-knowledge and situational awareness (enhancing self-reflection), and 78% (*n* = 147) stated that they enjoyed writing both exercises. 

### 3.1. Feelings during the First Day of Confinement

The feelings expressed by the students regarding the first day of confinement were grouped into 21 categories of which the 5 with the highest and lowest frequencies were uncertainty with 40.4%, sadness-worry with 29.7%, bewilderment with 28.7%, disbelief with 26.2%, and fear with 20.7%. The least frequent were dependence with 2%, anger with 2.5%, helplessness with 3.5%, surprise with 5%, and expectation with 5.5% (Figure 1). Narrative descriptions, in general, synthesise feelings in a more prosaic or superficial way, while these are expressed in a more intense and penetrating way in poems. The same is repeated in the narratives and poems written in the sociopoetics section but focused on their academic interests and uncertain professional future. 

#### 3.1.1. Threshold Phase (NBN: Identification of the Narrative Experience during the First Day of Confinement)

In this phase, the students described their thoughts about this situation. Uncertainty and insecurity stand out as the central ideas of the experiences provoked by confinement (NBN3). Students also reflected sadness in their narrative (NBN7). Expressions of fear associated with the above feelings were also frequent (NBN11). Disbelief or difficulty in assimilating what was happening was another of the most frequent themes in the narratives (NBN 10, 15, 27, 38, 59, 88, 132, 166). Helplessness and anger were recurrent themes in the narratives (NBN 45, 60, 79, 89, 89, 103, 127, 178). Among the reasons for anger, the feeling of manipulation of information is noteworthy (NBN169). In some cases, students tried to neutralise the worry and anxiety of the moment with the relief of being at home (apparently safe) (NBN, 169, 173). Furthermore, sadness and fear were countered with some hope (NBN89).

#### 3.1.2. Transthreshold Phase (PC: Expression of Feelings during the First Day of Confinement)

In this short selection of poems, the students’ feelings are described:

PC3 I accept the confusion/the uncertainty/the fear/and the emotional ups and downs/because that is the price to overcome/this overwhelming situation.

PC7 What will happen I ask myself day by day/ an unknown virus attacks with cruelty/it was a flu or so they said, with time we will see its lethality/I am hopeful, we are strong, a gust of wind will not knock us down/ strong and united we will win/tomorrow the dawn will come again.

PC10 Isolation plunges me into loneliness/ from which I can’t escape with social networks/ everyone tells what they want without rhyme or reason/but no one really knows what’s going on/I’m saddened by the insecurity caused by so much uncertainty. 

PC15 First day, rising sun/We don’t understand what’s going on/We don’t understand what will happen/Only fear between the curtains/Impotence of the living room/You struggle with every pillow.

PC32 And suddenly the war comes/And you feel weak and helpless that you can’t help/That your best contribution is to do nothing/Stay at home and watch the time go by.

PC45 Many days ahead/Not sure what will happen/At the moment I make my plans/Thinking that I’ll get out of this.

POC59 Today, 14 March, we lock ourselves in/We can’t see each other any more/Not knowing when we’ll see each other/Not knowing when we’ll be free.

PC81 It arrived, surprise and fear/ but also joy/to know that there would be rest, relief/ to make up for the time.

PC141 Another day locked up/ Another day without being able to be with friends/Another day without knowing what is going to happen/Another day to think that this ruin is going to pass.

PC169 All the information is manipulated/It’s sad to see the falsehood/Nobody knows, no/Nobody knows what’s going to happen/But they reassure us with children’s stories/Bad stories for children.

PC173 No one expected/The extent that this would have/To have to stay at home/Morning, night and day.

### 3.2. Feelings during the Most Significant Day of Confinement

On their most significant day, 22 causal feelings were identified (Figure 2). The students mostly considered the most significant day 33% (*n =* 66) to be the one with the highest emotional intensity that provoked an “explosion of feelings”, that is, emotionally charged conflicting feelings: The celebration of Mother’s Day, the birthday of a family member or partner, mourning the death of a loved one, etc. (*n =* 66); the day that caused an “explosion of feelings”; the day that caused them great anxiety and concern for the health of a family member or relative diagnosed or admitted to hospital, 21.7% (*n =* 43). Another significant motivation for the day was the “change of phase”, 20.7% (*n =* 41), which provoked feelings of joy, but also the fear of return. Anger and surprise at inappropriate behaviour such as the celebration of parties or drinking bouts, especially among young people, provoked a great emotional impact in 20.6% (*n =* 39). Mistrust and insecurity were other feelings that surfaced on a particularly worrying day, which together with hopelessness constituted 19.2% (*n =* 38); reunions with friends and family, the decrease in the number of deaths, the applause of solidarity from balconies, and new births during confinement constituted an emotional conglomerate for 15.6% (*n =* 31).

#### 3.2.1. Threshold Phase (NBN: Identification of the Narrative Experience during the Most Significant Day of Confinement)

In this phase, the students reflected on the greatest emotional impacts and wrote about them, reflecting on their possible causes: The death or illness of a loved one (friend or relative): NBN3, NBN7, NBN56; NBN83, NBN, 102. Moreover, they noted the day they learned that the number of deaths from the pandemic in the country had exceeded a significant figure (NBN 173, 193). The applause from balconies touched many students, making it their most significant day (NBN32, 37, 77, 89, 107, 133, 189). For others, the day of returning home from university or hospital was the most special day (NBN45, NBN193).

#### 3.2.2. Transthreshold Phase (POC: Expression of Feelings during the Most Significant Day of Confinement

PC3 Death is just a silence/without past/without mould/without smell/It’s a thick fog/that gets in the eyes/that destroys the voice/and corners one for good.

PC7 Fear/A powerful feeling/Swept everything away/Leaving us desolate and with our hearts in a fist.

PC32 How valuable it is to take care of the one who takes care of you/They at the foot of the cannon/With a thousand obstacles and facing all/Without armour and without swords/Saving the innocent and killing the bad.

PC45 Excitement for the return home/And many things to tell/My family I’ll be able to embrace/But locked up we’ll have to stay.

PC173 I’m afraid to see/How so many people are leaving/But I’ve noticed that I’m even angrier/At not being able to help.

### 3.3. Feelings on the Last Day of Confinement

On the last day of confinement, 15 feelings were identified that provoked an “aesthetic waterfall” in the students (Figure 3). These included the relief experienced at the prospect of finally returning to routine 55.5% (*n =* 110); in parallel, other types of feelings emerged, such as fear resurfacing in 53% (*n =* 105); others, as if they could not believe it was all over, experienced a feeling of sustained hopelessness (projecting from the immediate past of confinement) (47.4%, *n =* 94). Another group also had mixed feelings (joy and sadness; fear and anger simultaneously) (35.8%, *n =* 71); negative feelings such as boredom and anger also abounded and together accounted for 64% (*n =* 127).

#### 3.3.1. Threshold Phase (EBN: Identification of the Experience during the Last Day of Confinement)

For many, the last day was the beginning of a return to normality: Doing sport (NBN3, 17, 32, 34, 56, 77, 98, 172) and reconnecting with family (NBN7, 90, 107). Some found it hard to believe that it was all over and had mixed feelings: Joy that the pandemic seemed to be ending and fear of a resurgence (NBN 34, 77, 173). Finally, some students felt that confinement would be the beginning of a new era both socially and personally (NBN45, 78, 131).

#### 3.3.2. Transthreshold Phase (PC: Expression of Feelings during the Last Day of Confinement) Threshold Phase (EBN: Identification of the Experience during the last Day of Confinement)

PC3 “Don’t give up, you’re still in time/To reach out and start again/Accept your shadows/Enter your fears/Return to flight/Don’t give up, that’s what life is all about/Continue the journey/Continue the journey/Follow your dreams/Cross time/Run the rubble/And uncover the sky/Don’t give up/Please don’t give in/Even if fear bites/Even if the sun goes down/And the wind dies/There’s still life in your dreams (…)”.

PC7 If there were a single word/a single word/ that came close to it…/it would be surrealism.

PC32 And it’s in difficult times that one learns to value/Bailate life they said/And here I am like a little bird in a cage/With atrophied wings/Waiting for them to open the door/To go out and fly.

PC45 Many days have passed/Though we still don’t see the end/This has brought about a change/That will be crucial in our life.

PC173: Some of us from this confinement/We will leave having grown/And others surely/Will learn little.

## 4. Discussion

These feelings experienced by the students in a new situation (the field shaped by the pandemic), as Bordieu and Wacquant [8,9] point out, are a consequence of the previous matrix of schemas and perceptions with which they face an unexpected, different, unthinkable situation. The previous experiences constitute the root of the students’ emergent feelings during the pandemic (field), and the subjectivity of feelings in a shared context (confinement) is transformed into intersubjectivity [8]. In this intersubjective process, the students constructed their feelings socially [6,7]. For example, the feeling of uncertainty is so pervasive because, in reality, all students have shared a common prior context that encouraged their expectations, and as this context vanished amidst the chaos generated by the pandemic, intersubjectivity transformed the phenomenon into an objectifiable one. Resilience manifested in the need to learn from the situation (self-learning) may be rooted in previous experiences where positive feelings have prevailed and stimulated proactive behaviours (POC3, POC32, POC45, POC173) [3,14]. 

As Heller points out, students have experienced emotions that, while involved in the pandemic, have been transformed into feelings. In the course of the readings/auditions, the students reported feeling some relief in writing poems in which they unloaded the feelings experienced during stressful situations such as fear, isolation, loneliness, bewilderment, and uncertainty (POC3, 7 POC45) [5]. In this sense, writing functioned as therapy, provoking catharsis through which the students purged their tensions and chased away their demons in the face of the feelings provoked by the death of loved ones (PC 3, 23, 33, 67, 87, 92, 101) [16,17,18].

In the discussion during the readings, the students noted that the great variety, both in terms of categories and the expression of feelings, is linked to previous experiences. These pre-confinement experiences function as experiential revelation triggered by reflection, self-reflection, and heteroreflection through NBN and PC (NBN and PC 23, 72, 93, 101, 133, 168, 173) [21,22,23,24]. BNN and PC also contribute to filtering emotional and behavioural responses even in a phenomenon as disruptive as the pandemic [12]. 

In the same vein, Huerta argues that if feelings were exclusively innate, all individuals would be affected in the same way by the outbreak of a phenomenon [7]. On the other hand, the results show that the aesthetic study (poetry of care) facilitates introspection by establishing a window into the therapeutic experience (PC 32, 44, 56, 79, 139) [26], constituting a source of knowledge and self-knowledge in an unlikely situation [12,19,26]. Likewise, PC facilitates the valuation of the most essential and transcendent of the phenomenon by manifesting the students’ authenticity or “truth” regarding confinement and the COVID-19 pandemic [11]. It should be noted how students have reorganized their hierarchy of needs as a consequence of the experiences during confinement (NBN and PC 3, 6, 45, 78, 98, 101, 132, 173) [14,15]. 

The pandemic has provoked a rupture of stability or equilibrium with respect to students’ daily lives inducing unsuspected and surprising cultural and social practices (which have manifested themselves through a highly polarized aesthetic reaction in feelings). From the perspective of habitus [8], some of the most frequent feelings in students, uncertainty, bewilderment, and disbelief, show the difficulty they have in understanding a reality that has broken their schemes (NBN 45, 59) forged in a society whose references are linked to social and educational structures that have shaped their values, beliefs, and expectations within the framework of a society characterized by welfare, scientific technological development, democracy (family, school, university, political system), and ethical considerations [1,11]. Some students surprised themselves in the writing process by identifying feelings that, although they had always been part of them, they had not been aware of. Perhaps this is related to the breakneck speed of these times that facilitates a lack of focus on transcendence and personal truth [6,11,23].

The unawareness of feelings that students had is evident in narratives such as this one: “I didn’t know I had that feeling inside and as I have been writing the poem it has been coming out” (EBN144) [4,5]. Perhaps more relevant is the use of poetry when the feelings are so intense and dense that in their complexity are amalgamated pain, love, and compassion; the sublime and the beautiful of care [1].

Some of the most widespread feelings, uncertainty and bewilderment, have been identified by some students as synonyms for emptiness, silence, fear, and dependence. This is a consequence of dialectical paralysis caused by the unusualness of a confining context that has made it difficult for students to choose words to express what is happening and, above all, what is going to happen [27]. 

To a lesser extent, students express helplessness associated with feelings of anger, especially those who work in healthcare settings as assistants or orderlies. Others express contrasting feelings denoting (PC32, 123) concern but also some relief at being sheltered with their family at home or express aspects such as fear, uncertainty, and sadness, but with an optimistic outlook based on hope.

The first consequence of writing poems, as a creative action, has been the activation of their imagination and intuition [15]. The practice of writing—narrative and poetic—is inherent to critical reflection and self-reflection [23] through which the student re-encounters himself by understanding himself in a new situation to which, as Ricoeur states, he is able to assign a new meaning [13]. This is manifested in the opposing feelings that reveal, on the one hand, the vision of the complexity of the phenomenon, and, on the other hand, a struggle to find oneself again by establishing an identity space between the negative and positive aspects caused by the crisis (E89, E141). There is also an abundance of expressions associating feelings of sadness and uncertainty with critical reflection [15] regarding the media, information, and management of the pandemic (E169). 

### Limitations of the Study

There is a difference between people’s aesthetic sensitivity and creativity, so it is convenient to establish a suitable environment that stimulates students’ participation by explaining the nature of the work and establishing careful planning.

It is important to clarify that this is not a literary work and that, therefore, it is not necessary to force the search for rhyme or poetic effects typical of literary activity. However, some students concentrated on the structure of the poem and became preoccupied with the length and the need for their poem to rhyme. This distracts students from the process as a reflective activity and is a finding shared by similar studies [24]. However, engaging in the task of writing a poem supports the development of writing skills, language, and vocabulary [28], and even encourages the reflective process.

## 5. Conclusions

It is necessary to start from reality: There is no tradition in the use of these methodologies—or other methodologies focused on creative thinking—in the nursing community, so it is advisable to apply strategies that familiarize students and nursing professionals with them.

The hypothesis of the study has been confirmed: The combination of Narrative-Based Nursing and poetry of care is a tool that contributes to self-knowledge and awareness of students’ experiences and feelings during the confinement caused by the COVID-19 pandemic.

The research question has been adequately answered by identifying the feelings experienced by students during the COVID-19 pandemic.

One of the strategies used to facilitate the writing of the poems was a preliminary phase in which the students described their feelings during the confinement in prose.

The proposed objectives have been achieved by identifying the feelings experienced by the nursing students during the beginning of the confinement during the COVID-19 pandemic: Uncertainty, bewilderment, disbelief, and surprise (due to the unexpected), feelings of social-physiological imbalance and fear, all summed up in one word: “surrealism”.

Narrative-based nursing and poetry of care have proven their relevance in the process of identifying students’ feelings both subjectively (individual) and intersubjectively (group).

Students have expressed their feelings as a reflection of the impact caused by the pandemic (social, physiological, and psychological imbalance) on their mechanism of satisfaction of needs in everyday life and on the process of the social construction of feelings. This loss of homeostasis externally driven by COVID-19 (field) has impacted students’ values, beliefs, traditions, and knowledge (habitus).

The use of narrative-based nursing and poetry of care is producing a revolution in clinical practice and the daily life of students. The assessment and analysis of feelings and emotions that occur in the clinical context during nurse–patient interactions have practical implications for nursing: It contributes to the optimization of communication between professionals and patients and facilitates knowledge about the aesthetic causes of burnout when feelings are not assimilated, producing stress, anxiety, and emotional fatigue. It also facilitates practical reflection on experiences that affect the way in which nursing is studied and interpreted and critical thinking and practical reflection and has very relevant therapeutic potential in the clinical and pandemic context. Finally, through narrative-based Nursing and poetry of care, students reflect and develop consciousness about the impact of COVID-19 on nursing.

This type of study has an impact on nursing education policies by demonstrating the importance of NBN and CP for the analysis of feelings and emotions. The integration of NBN and PC in nursing curricula at an international level is causing a substantial change in the way nursing is understood and practised. This integration of an aesthetic dimension in the studies will facilitate changes that will have an impact both on the humanisation of care and the consideration of feelings and emotions in problems such as demotivation, stress, and burnout for health professionals.

This study may contribute in the future to the development of qualitative research focusing on NBN and CP, especially in the clinical context. The results of these studies will contribute to the humanization of care. Furthermore, this research should have an impact on the integration of narrative methodologies in the curricula of nursing faculties.

## Figures and Tables

**Figure 1 ijerph-19-13919-f001:**
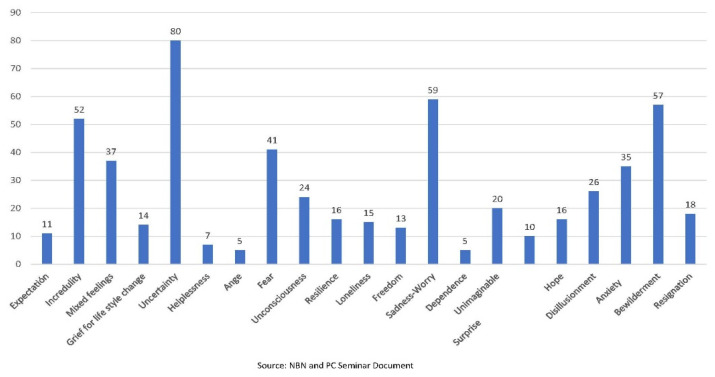
Feelings during the first day of confinement.

**Figure 2 ijerph-19-13919-f002:**
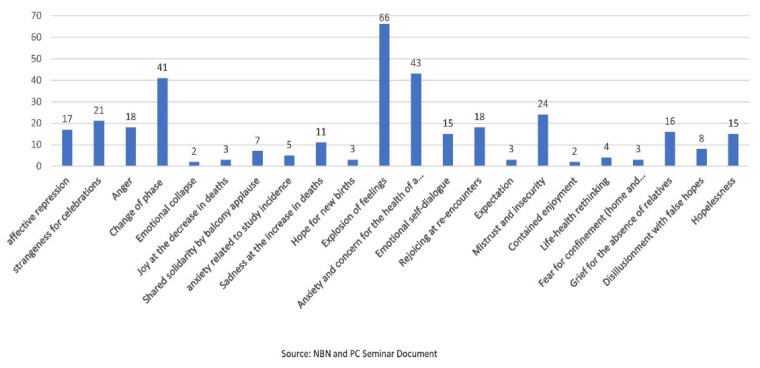
Feelings during the most significant day of confinement.

**Figure 3 ijerph-19-13919-f003:**
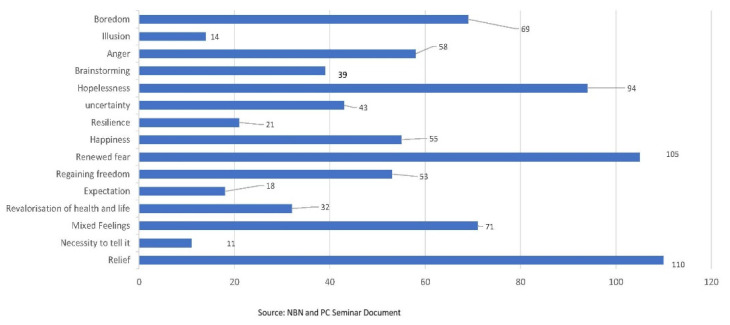
Feelings on the last confinement day.

## Data Availability

Not applicable.

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
