# Peer review of "The Feelings of Nursing Students during the COVID-19 Confinement: Narrative-Based Nursing and Poetry-of-Care Perspectives"

_ijerph, 2022, doi:10.3390/ijerph192113919_

Round 1

Reviewer 1 Report

I have some problems with this text;

1.) I cannot find a clear research question or a clear hypothesis formulated in the introduction.

2.) "PC" and "NBN" are used in the introduction, but they have not been   introduced.

3.) The introduction ends with a kind of recommendation on how to write a research paper ( line 69- line 78). This is quite strange. The same is true for "4.1 Limitations of the study" (line 376- line 379).

4.) An explanation of the methods used is missing.

 5.) It is difficult to understand the "discussion" and the "conclusion" as long as it is not clear what questions should be answered or what kind of hypotheses should be tested.

Author Response

Point 1: I cannot find a clear research question or a clear hypothesis formulated in the introduction.

Response 1: The authors wrote the research hypothesis and the research question in the introduction (p-2/79-83).

Point 2: "PC" and "NBN" are used in the introduction, but they have not been   introduced.

Response 2: The authors describe the concepts of CP and NBN in the introduction (p-2/69-74)

Point 3: The introduction ends with a kind of recommendation on how to write a research paper ( line 69- line 78). This is quite strange. The same is true for "4.1 Limitations of the study" (line 376- line 379).

Response 3: The authors deleted both paragraphs.

Point 4: An explanation of the methods used is missing.

Response 4: The authors described the method applied (p-2/95-100; p-3/101-104).

Point 5: It is difficult to understand the "discussion" and the "conclusion" as long as it is not clear what questions should be answered or what kind of hypotheses should be tested.

Response 5:  The authors clarified the impact of the hypothesis and the research question on the results. on the results of the study. (11/408-413).

Reviewer 2 Report

The paper does not contain a statistical analysis

Where is the study assignment?

Where are the limits of the study?

Where did you get the sample and what is the population of the study?

The paper needs further revision

Author Response

Point 1: The paper  does not contain a statistical analysis.

Response 1: We are a research group specialising in qualitative research. The study is qualitative with narrative sources and the evidence is based on subjectivity. It does not require any statistical treatment other than frequencies/percentages.

Point 2: Where is the study assignment?

Response 2: The study was carried out on first year students of the nursing degree at the University of Alicante (p-3/124-126). The assignment to collect the data is described in the Methods section (p-3/128136).

Point 3: Where are the limits of the study?

Response 3: The limits of the study are described in the study limitations section ( p-11/385-396).

Point 4: Where did you get the sample and what is the population of the study?

Response 4: This paragraph describes the context in which the sample was taken (p-3/123-127). The sample, in this case, coincides with the population.

Reviewer 3 Report

This manuscript is a qualitative study of NBN and PC, and the author analyzed the impact of COVID-19 on nursing, which has a positive significance. The following points need to be improved.

1. In Section I, the author should add some introduction about pandemic research which related to this study;

2. As a research on human beings, discussions on ethical issues should be added to the paper;

3. The prospect of further research should be briefly outlined in the Conclution.

Author Response

Point 1. In Section I, the author should add some introduction about pandemic research which related to this study;

Response 1: The authors cited articles in the introduction describing the characteristics of disruptive contexts such as pandemics and their impact on feelings (p-2/61-62).

Likewise, we cite articles in which the field is equated with the disruptive (pandemic) context that affects the socialisation of feelings (habitus) (in the methodological section and also in the discussion (p-3/114-116; p-9/306-319).

Point 2: As a research on human beings, discussions on ethical issues should be added to the paper.

Response 2: The ethical characteristics of the NBN and PC are cited in the introduction (p-2/58-61); (p-3/146-149); in the methodological section (p-3/146-149) and also in the discussion (p-10/353-354)

Point 3: The prospect of further research should be briefly outlined in the Conclution.

Response 3:  The authors describe the impact of this study on the development of future research (in the conclusions) (p-12/449-452).